# Seasonal Drought Prediction for Semiarid Northeast Brazil: Verification of Six Hydro-Meteorological Forecast Products

José Miguel Delgado[1], Sebastian Voss[1], Gerd Bürger[1], Klaus Vormoor[1], Aline Murawski[2], José Marcelo Rodrigues Pereira[3], Eduardo Martins[3], Francisco Vasconcelos Júnior[3], and Till Francke[1]

[1]Institute of Earth and Environmental Sciences, University of Potsdam, Germany
[2]German Research Centre of Geosciences GFZ Potsdam, Germany
[3]Research Institute for Meteorology and Water Resources - FUNCEME, Fortaleza, Brazil

*Correspondence to:* José Miguel Delgado (martinsd@uni-potsdam.de)

**Abstract.** A set of seasonal drought forecast models was assessed and verified for the Jaguaribe River in semiarid northeast Brazil. Meteorological seasonal forecasts were provided by the operational forecasting system used at FUNCEME (Ceará's research foundation for meteorology) and by the European Centre for Medium-Range Weather Forecasts (ECMWF). Three downscaling approaches (empirical quantile mapping, extended downscaling and weather pattern classification) were tested and combined with the models in hindcast mode for the period 1981 to 2014. The forecast issue time was January and the forecast period was January to June. Hydrological drought indices were obtained by fitting a multivariate linear regression to observations. In short, it was possible to obtain forecasts for *a)* monthly precipitation, *b)* meteorological drought indices, and *c)* hydrological drought indices.

The skill of the forecasting systems was evaluated with regard to root mean square error (RMSE), the Brier skill score (BSS) and the relative operating characteristic skill score (ROCSS). The tested forecasting products showed similar performance in the analyzed metrics. Forecasts of monthly precipitation had little or no skill considering RMSE and mostly no skill with BSS . A similar picture was seen when forecasting meteorological drought indices: low skill regarding RMSE and BSS and significant skill when discriminating hit rate and false alarm rate given by the ROCSS (forecasting drought events of e.g. $SPEI_1$ showed a ROCSS of around 0.5). Regarding the temporal variation of the forecast skill of the meteorological indeces, it was greatest for April, when compared to the remaining months of the rainy season,while the skill of reservoir volume forecasts decreased with lead time.

This work showed that a multi-model ensemble can forecast drought events of time scales relevant to water managers in northeast Brazil with skill. But no or little skill could be found in the forecasts of monthly precipitation or drought indices of lower scales, like $SPI_1$. Both this work and those here revisited showed that major steps forward are needed in forecasting the rainy season in northeast Brazil.

## 1 Introduction

Northeastern Brazil has historically been the epicenter of major drought events. Fioreze et al. (2012) identified 100 severe droughts since the 16th century in this region, while Marengo et al. (2016) identified 68 major events for the same period.

Within this region, the state of Ceará has been in the frontline of the fight against this natural hazard. This has been both due to the impacts suffered in the past and to the measures taken to improve its resilience.

Droughts in Ceará reflect a meteorological anomaly over the tropical Atlantic Ocean. Dry years are generally related to a positive sea surface temperature (SST) anomaly on the tropical North Atlantic, associated with a negative anomaly on the tropical South Atlantic and over the equator. This forces a northward shift of the intertropical convergence zone, taking the rainbelt to northern latitudes. The causes for this anomaly are linked to the occurrence of the El Niño Southern Oscillation and to the North Atlantic Oscillation (Hastenrath, 2012).

Past famines and mass migrations triggered large investments in infrastructure in recent decades. These investments brought hundreds of strategic reservoirs and thousands of small dams to a semi-arid landscape, which are being managed according to a transparent water allocation process (Formiga-Johnsson and Kemper, 2005). In order to support water allocation and management, the state runs a seasonal drought forecasting system and issues annual quantitative and qualitative forecasts of the magnitude of the rainy season. These predictions can support decisions ranging from agricultural management (choice of crop, planning of seeding time) to water distribution and reservoir operation.

Currently, the forecasting system in Ceará is based on the general circulation model ECHAM4.6 (Roeckner et al., 1992). It runs from January to August on persisted SSTs (observed SST anomalies which are assumed invariant), covering each year's rainy season (February to April). The forecasts produced by this model are generally downscaled with the NCEP regional spectral model (Juang et al., 1997), in order to resolve the spatial variability of Ceará. Verification and the current forecast can be retrieved under http://www3.funceme.br/previsao-climatica/. For downscaled forecasts check http://seca-vista.geo.uni-potsdam.de/.

In this study we intend to evaluate and extend this prediction system by employing

1. an additional underlying GCM,

2. a statistical approach based on the classification of weather patterns,

3. empirical-statistical downscaling methods to increase the spatial resolution and temporal fidelity of the predictions, and

4. drought indices as powerful integrative descriptors for the description of drought severity.

By these means, we aim to address the following questions:

What skill do the seasonal *meteorological drought* forecasts have?

While the term *meteorological drought* focuses on the atmospheric forcing causing water shortage, its effective implications for society are more specifically accounted for by the term *hydrological drought* (Araújo and Bronstert, 2016). Since the aim of the prediction system is to support water management, we sharpen the previous question in this regard:

Can we forecast *hydrological* droughts in Ceará based on these seasonal meteorological forecasts?

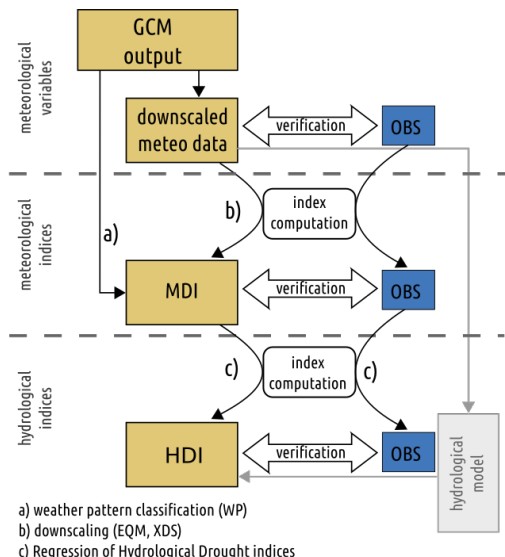

**Figure 1.** Flowchart explaining the methodology used for predicting meteorological data, meteorological drought indices (MDI) and hydrological drought indices (HDI).

## 2 Methods

### 2.1 General approach

This work employed a cascade of models and algorithms ranging from two general circulation models (one atmospheric and one coupled) at the top to hydrological indices at the bottom (Fig. 1). Each step involved different types of target variables being

forecasted: The *meteorological forecasts* (Fig. 1, top) refer to meteorological variables ("meteo data") from GCM-forecasts and the subsequent downscaling and bias correction to match the spatial and temporal resolution. The *meteorological indices* (same figure, centre), refer to the indices that were used to describe the magnitude of the forecasted meteorological drought. Finally, the *hydrological indices* (same figure, bottom) were calculated based on meteorological indices in an attempt to infer the magnitude of a hydrological drought characterized by meteorological and hydrological properties. To allow for the comparison

with observations, we use results of GCM hindcast, i.e. a model that has been run with data only known until the specified time in the past. As these are supposed to represent and technically resemble true forecasts, they are referred to as "forecasts" henceforward. All results and computations after the statistical downscaling have a monthly time step. Similarly, all results and computations here presented were aggregated to selected subbasins (Fig. 2).

### 2.2 Study area

The spatial domain chosen for this analysis is the Jaguaribe river basin. Due to the river's regional importance, a lot has been written about its hydrology and development (see e.g. Araújo and Bronstert, 2016; Araújo, 1990). The Jaguaribe is the

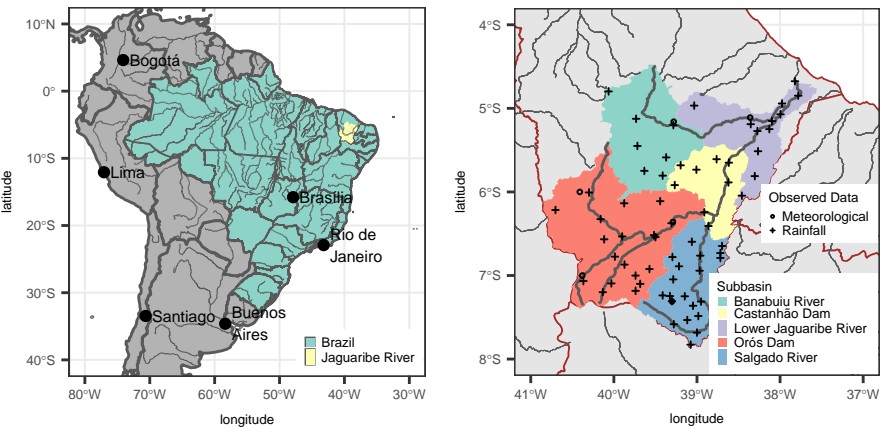

**Figure 2.** Left panel shows the location of the Jaguaribe River basin in South America. Right panel shows the Jaguaribe River together with its main tributaries, division into subcatchments used in this work, and meteorological and rainfall observations stations.

most important river in Ceará. Its catchment has an area of $70000\,\mathrm{km^2}$ and is home to about 2.7 million people (IPECE, 2017). Annual precipitation amounts to $755\,\mathrm{mm}$, of which about 90% fall in the months January to June. The rainfall season comprising the months January to May is often considered key in securing water reserves for the whole year. June contributes with considerably less rainfall.

Average potential evapotranspiration is estimated to $2100\,\mathrm{mm}$. Due to its dominant geology composed of a crystalline complex, aquifers in the region are unproductive. Runoff is practically the only source of drinking water for people and animals as well as irrigation. To that end, most of the water is stored in thousands of reservoirs of all scales across the watershed.

    The main tributaries are the Banabuiú river in central Ceará and the Salgado river in southern Ceará. We aggregated the results of this research into five subcatchments: the aforementioned tributaries Banabuiú and Salgado, the upper (upstream of

Orós Reservoir), middle (upstream of Castanhão Reservoir) and lower (downstream of Castanhão Reservoir) Jaguaribe. An overview of the state and location of these catchments and tributaries is given in Fig. 2.

## 2.3   Seasonal Forecast Models ("GCM output")

To address the first research question we employed different combinations of dynamical and statistical models and a weather pattern classification methodology to produce meteorological drought indices. The dynamical seasonal forecast models were

provided by FUNCEME and ECMWF in the form of hindcasts for the period 1981 to 2014. Details like resolution, reference and short description are given in Table 1.

    The ECMWF operational seasonal forecasting system S4 has 51 ensemble members and six months lead time. It is a fully coupled atmosphere-ocean model. The system has been systematically verified (Vitart, 2013; Molteni et al., 2011; Richardson et al., 2012). The hindcast version of the system has the same specifications of the operational model but only 15 ensemble

members. It is available for academic purposes and is here employed as a benchmark for the verification of the regional forecasting system in operation in Ceará.

The seasonal forecasting system implemented at FUNCEME (Ceará's hydro-meteorological agency) is based on the general circulation model ECHAM4.6. Details on this model can be found in Table 1. The operational and hindcast version have 20 ensemble members and are run on initial sea surface temperature (SST) anomalies persisted during the forecasting period (8 months). The initial state represents a typical (but random) realization of late December as derived from AMIP-type runs (Gates et al., 1999) The AMIP run starts in 1961 and is forced by monthly observed SSTs (NOAA Optimum Interpolation SST V2). Therefore, potential forecast skill is solely based on oceanic memory. The forecasting system of FUNCEME is in operational use and seasonal forecasts are released monthly.

## 2.4   Downscaling of GCM output

In order to predict precipitation over particular locations it is necessary to downscale the GCM forecasts. Three statistical downscaling approaches were employed: expanded downscaling (XDS), empirical quantile mapping (EQM) and weather pattern classification (WP, see Table 1 for details and references). To differenciate between two fundamentally different downscaling approaches, weather pattern classification will not be refered to as downscaling approach/method throughout the text.

The downscaling approaches used here yielded a full set of meteorological variables distributed across the catchment at points where observations were available (daily mean temperature, relative humidity, wind speed and daily total precipitation and radiation). The forecasting products obtained from the combinations of GCM and downscaling will be named after their components: *XDS:ECHAM*, *XDS:ECMWF*, *EQM:ECHAM*, *EQM:ECMWF*, *WP:ECHAM*, and *WP:ECMWF*.

Weather patterns were classified using the SANDRA methodology described in Philipp et al. (2016). The selection of the *optimal* classification was done visually in respect to the explained variation of the observed meteorological drought indices. The classification itself was independent of the MDIs, so that no artificial skill was to be expected from forecasting the stations. Only MDI scales of 1, 12 and 36 months were calculated.

## 2.5   Drought Quantification using Drought Indices

Meteorological droughts were quantified in magnitude and temporal scale using meteorological drought indices (MDI). After careful appraisal regarding data demand and current conventions, the following indices were selected: $SPI_1$, $SPI_3$, $SPI_6$ (Svoboda et al., 2012; McKee et al., 1993), $SPEI_1$, $SPEI_3$ and $SPEI_6$ (Vicente-Serrano et al., 2009). The subscripted numbers (e.g. $SPI_1$) refer to the temporal scale in months for which the index was computed.

The forecast is generated at the beginning of January for the period from January until June. Indices obtained by downscaling forecasts with temporal scale greater than the lead time of the forecast will include values from the observation set. $SPI_6$, for example, will contain five month of measured precipitation in the January forecast. In June, the same index will be calculated exclusively with forecasted precipitation. The skill of a $SPI_6$ forecast for some months is therefore expected to be greater than the skill of a $SPI_1$ forecast beforehand. This feature does not apply to WP classification.

**Table 1.** Output variables of each prediction model used in this paper

| Model/Method | Short description | Reference | Spatial Scale |
|---|---|---|---|
| FUNCEME Seasonal Forecast System | A 20 member ECHAM4.6 ensemble. Atmospheric circulation model, initial SSTs persisted for 6 months. Initial conditions of the atmosphere modeled by AMIP-type run (starting in 1961). AMIP run is forced by monthly observed SSTs (NOAA Optimum Interpolation SST V2). | Roeckner et al. (1992) | approx. 2.8 degree longitude/latitude |
| ECMWF Seasonal Forecast System | A multi-model 15 member ensemble including ocean circulation. Initial conditions coming from ERA Interim. | e.g. Stockdale et al. (1998) | approx. 0.7 degree latitude/longitude |
| expanded downscaling | Simulates local events consistent with prevailing atmospheric circulation while preserving local covariability | Bürger (1996) | network of monitoring stations |
| empirical quantile mapping | Improves systematic biases throughout the statistical distribution by mapping the empirical cumulative distributions of the observed and modelled variable | e.g. Wetterhall et al. (2012) | network of monitoring stations |
| weather pattern classification | Including pre-selection of variables, variable combinations and spatial domain. | e.g. Murawski et al. (2016); Philipp et al. (2007) | network of monitoring stations |

Time scales greater than 6 months are of no value for the verification of the forecasting system in terms of meteorology, as rainfall in the preceding dry season is usually negligible. However, the hydrology of Ceará is characterized by long-term memory introduced by a vast network of reservoirs. Additionally, drought events in this region are known to be long and creeping phenomena that must be quantified on large temporal scales. MDIs with long temporal scales will therefore have to be considered when designing the hydrological drought index (HDI) forecast model in the next section. To that end, we will employ shorter time scales for MDI verification, but keep longer time scales (greater than 6 months) in the regression of hydrological drought indeces (HDIs), since they provide a better fit for the forecast model.

Regarding hydrological droughts, various hydrological drought indices (HDI) were reviewed and two were considered suitable for this work. All other indices either a) require consumptive data for water use, which is impractical for the given settings or b) focus on streamflow, which misses the most important features (ephemeral rivers, role of reservoirs) of the hydrological

system of Ceará and many other semi-arid regions. The only index chosen from the literature was the surface water supply index (SWSI) as formulated in Doesken et al. (1991) with a weight of 0.5 for precipitation within the reservoir catchment and 0.5 for reservoir volume:

$$\text{SWSI} = \frac{0.5P(rs) + 0.5P(pr) - 50}{12} \tag{1}$$

where $P(x)$ is the non-exceedance probability of $x$ based on available historical records of $x$, $rs$ is mean monthly reservoir storage in the respective catchment and $pr$ is the monthly precipitation averaged for the respective catchment. The second index, $V$, was defined as the regional reservoir volume at the end of each month relative to the total regional reservoir storage capacity.

In terms of event prediction, the event considered for the meteorological drought indeces in use in this work is "dry spells

of moderate to extreme magnitude", translated by values lower than or equal to $-1$ in the SPI/SPEI scale. For precipitation a threshold based on the 30th percentile of the series of observed monthly precipitation was used. The threshold for defining HDI drought events was based on the 30th percentile of the series of observed monthly HDI. The reason for using the 30th percentile was the classification used by the regional agencies to separate between a "dry", a "wet" (above the 70th percentile) and a "normal" year.

**2.6   Regression of Hydrological Drought indices**

Forecasts of hydrological drought indices were obtained by searching and fitting a multivariate regression model to observations of hydrological drought indices and reservoir volume changes. As candidate predictors, meteorological drought indeces of all temporal scales were used..

For predicting SWSI the multivariate linear regression was fit directly to the hydrological index. For the regional reservoir

volume, $V$, two different approaches were followed. With the first approach M1, the multivariate linear regression was fit directly to the values of $V$, analogous to SWSI. With M2, the second approach, the multivariate linear regression was first fit to the monthly changes of $V$. Then the predicted value of $V$ was calculated by adding its predicted monthly changes to the most recent measured value in December. The regional reservoir volume predicted by the two regression models M1 and M2 are denoted $V_{M1}$ and $V_{M2}$ respectively.

Model parsimony was enforced by predictor selection comprising a heuristic search for the best Akaike information criterion (AIC) under the constraint of checking the predictors for multi-collinearity. To eliminate multi-collinearity between predictors, correlated predictors were replaced by their ratios.

Possible forms of multilinear regression include predictors as denominators of fractions. This implies that these predictors must not take the value zero, in order to exclude the division by zero. To enforce this condition, the MDIs in question were

removed from the time-series when approaching zero, in particular values in $]-0.1, 0.1[$.

## 2.7   Forecast Verification

At each level of Fig. 1, a verification of the forecast was performed. Three metrics were employed: the root mean square error (RMSE), the relative operating characteristic skill score (ROCSS) and the Brier skill score (BSS) (Wilks, 2005). Root mean square error is a scalar accuracy measure applied to the realisations of the ensemble forecast. The Brier score is also a scalar ac-
curacy measure, though for verification of probabilistic forecasts of predefined events. The relative operating characteristic is a discrimination-based verification metric for forecasts of defined events. For more information on these metrics, we recommend chapter 8 of Wilks (2005)

RMSE was computed for each member, ensemble mean and climatology, i.e. long-term mean annual cycle. Climatology was considered the reference forecast. The mean square error was computed for monthly values in the forecast period (1981-2014,
January-June) and averaged over the entire period. The square root of this measure is the RMSE. It shows the capability of the model to correctly forecast monthly values, but it does not quantify the skill to predict particular events of water scarcity. January to June precipitation represents over 90% of the annual precipitation in the Jaguaribe basin.

Another important metric employed was the BSS. The Brier score can be seen as the sum of three terms *reliability*, *resolution* and *uncertainty*. The term *reliability* measures the differences between forecast probabilities and relative frequencies of the
observed event. Thus, low values of this score correspond to high reliability. *Resolution* measures the ability of the forecast to discern periods in which observed frequencies depart from average. Finally the term *uncertainty* quantifies the variability of the observations: when the event being forecasted almost never, or almost always happens, the uncertainty of the forecast is small. The Brier score is here understood as in Wilks (2005) as:

$$BS = \frac{1}{n}\sum_{k=1}^{n}(y_k - o_k)^2 \tag{2}$$

where $BS$ is the Brier score and $k$ denotes the index of the $n$ forecast-event pairs. $y_k$ is the forecast probability for each forecast-event pair $k$ and belongs to $[0,1]$. The forecast probability is calculated as the number of members of the ensemble that forecast an event divided by their total count. $o_k$ is the observation for each forecast-event pair, which can take the value 1 for an event and 0 when no event is observed in $k$.

The BSS is computed with respect to the Brier score of the reference forecast ($BS_{ref}$):

$$BSS = 1 - \frac{BS}{BS_{ref}} \tag{3}$$

and it can take any value lower than or equal to one. A forecast is said to have skill if its BSS is greater than zero.

The reference forecast was considered to be the climatological relative frequency of the predefined event. For example the climatological relative frequency for February is the number of times that a February observation, e.g. of precipitation, is considered an event divided by the total number of years in the hindcast period.
The last metric employed was the relative operating characteristic  skill score (ROCSS). The relative operating characteristic describes the ability to discriminate between true positives and false positives when forecasting a given event. It is normally

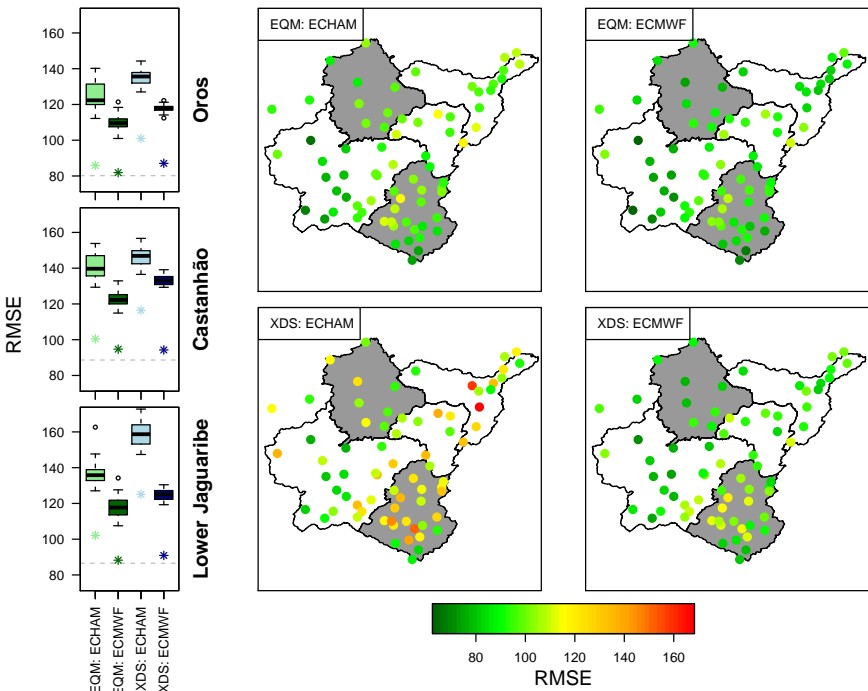

**Figure 3.** Root mean square error of the forecast of monthly precipitation. On the left panels, box plots show the spread of the RMSE of each model. The asterisks (*) show the RMSE of the ensemble mean. The RMSE of using climatology as a forecasting product is given by the grey dashed line. The four panels on the right show RMSE for each individual station for each model. Note that in general the ensemble mean ranks better than the best of the ensemble members.

calculated for a set of forecast probability bins, therefore having great importance for decision makers. ROCSS was calculated as

$$ROCSS = 2 \cdot AUC - 1 \tag{4}$$

as in Wilks (2005), where AUC is the area under the relative operating characteristic curve. The ROCSS can have values
5  between $-1$ and $1$, where anything below zero means *no skill*. A ROCSS of $0$ corresponds to the skill of a referrence random forecast.

## 3  Results and Discussion

### 3.1  Forecasting precipitation

The RMSE of the precipitation forecast are presented in Fig. 3. ECMWF ranks better than ECHAM, while EQM:ECMWF
10  results in the lowest RMSEs and XDS:ECHAM in the greatest. Still, the best results in terms of RMSE are comparable to the

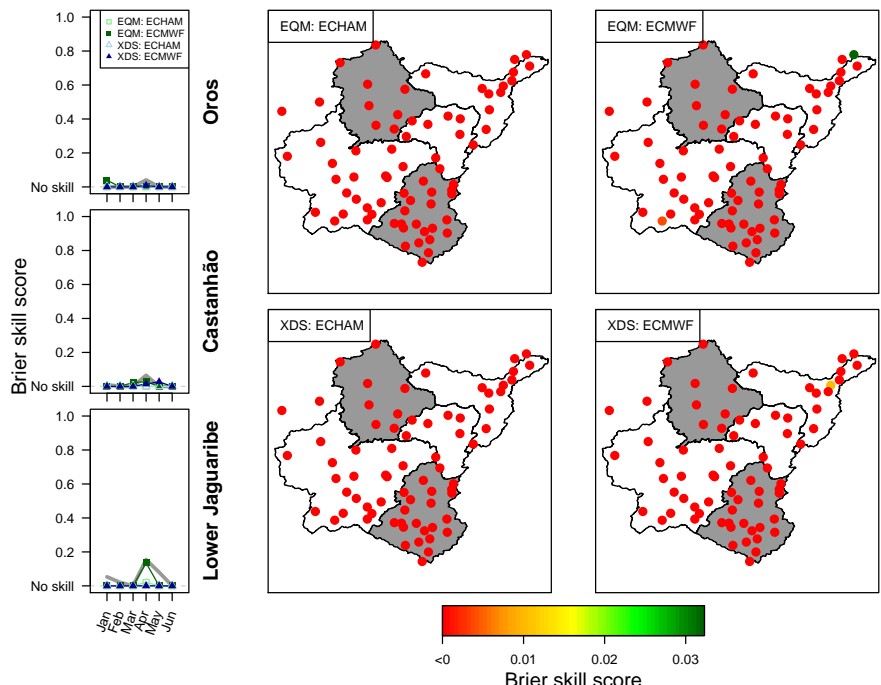

**Figure 4.** BSS of the forecast of a monthly low precipitation event. On the left side the BSS is shown for each model/downscaling combination and for the forecasting months averaged over the respective subcatchments. BSS values below zero were assigned a "no skill" category in order to improve readability. The grey line is the BSS of the multi-model ensemble. The four panels on the right show BSS averaged over all forecasting months for each station. Note that in most cases the forecast of monthly precipitation has no skill.

climatology, meaning that there is limited skill in forecasting monthly precipitation. The spatial distribution of RMSE of the ensemble mean in April shows a concentration of high RMSEs in the lower Jaguaribe catchment for EQM and in the Salgado catchment for XDS.

The ensemble mean of the forecast, shown by the asterisks in Fig. 3 as well as in other figures below, always displayed a
5  lower RMSE than *any of the ensemble members*. This happens because the ensemble mean "smoothes out unpredictable detail and presents the more predictable elements of the forecast" ((WMO) World Meteorological Organization, 2012). Despite its usefulness, the ensemble mean is not entirely appropriate for predicting drought events. Ensemble means do not provide any information on the probability of an extreme event.

Other than RMSE, which does not provide any information on the skill of event forecasts, the BSS is explicitly suited for
10  that purpose and is shown in Fig. 4. One remarkable observation is to be made regarding the BSS: skill is mostly absent when forecasting drought events based on precipitation and its thresholds. The only exception is the forecast for April, where the multi-model ensemble shows limited skill in the three regions considered. This results will be discussed further in face of the greater skill shown when forecasting drought events based on MDIs.

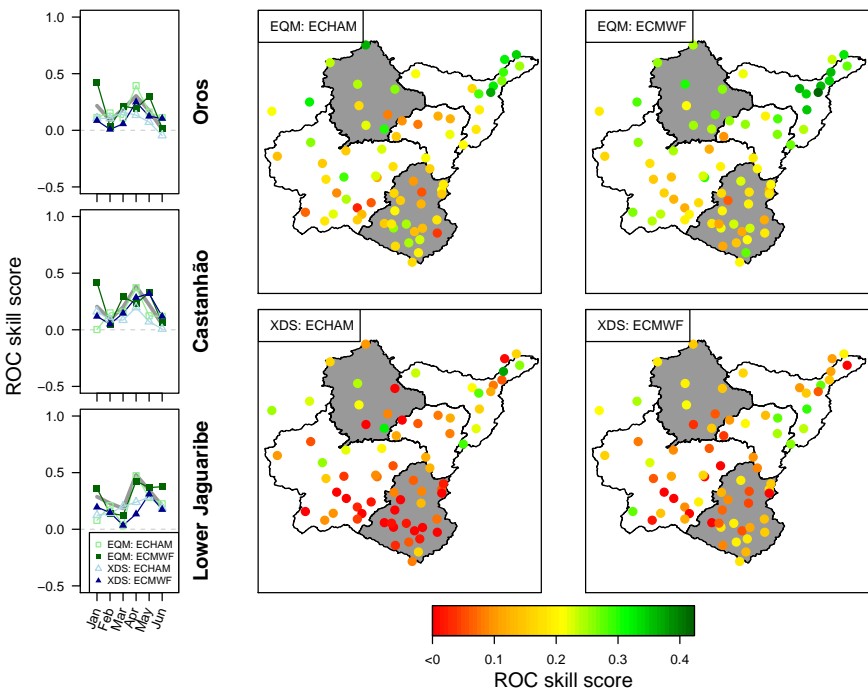

**Figure 5.** ROCSS of the forecast of a monthly low precipitation event. On the left side the ROCSS is shown for each model/downscaling combination and for the forecasting months averaged over the respective subcatchments. The four panels on the right show ROCSS averaged over all forecasting months for each station.

Still, all forecasting systems here presented show skill in discriminating events against false alarm forecasts. This is expressed by the ROC-curve shown in Fig. 5 . The variation of the ROCSS over time can be attributed to lead time (skill decreasing with increasing lead time) and to low or no precipitation in the months before and after the rainy season. Months of typically low precipitation showed poor ROCSS (Fig. 5: January, May, June). When comparing downscaling techniques and GCMs, EQM

5 mostly outperformed XDS, while the skill was less affected by using different GCMs.

To put our results into context, we could find three reports with a statement of verification concerning precipitation forecast in Ceará. Castro et al. (2013) presents a RMSE of between 120 and 130 mm for the *Sertão Interior de Inhamuns*, using an empirical model with forecasts issued in January for the period February to June. Moura and Hastenrath (2004), with a forecast issued in end of February for the period of March to June, i.e. with shorter lead-time than our work, shows a RMSE of 50 to

10 70 mm (Hastenrath and Greischar (1993) obtained similar results).

## 3.2 Forecasting meteorological drought indices

A time series of seasonal MDI forecasts was plotted to illustrate the forecast spread given by model EQM:ECMWF (Fig. 6). The improvement provided by the ensemble mean, when compared to each member, is clearly visible. Also visible are several

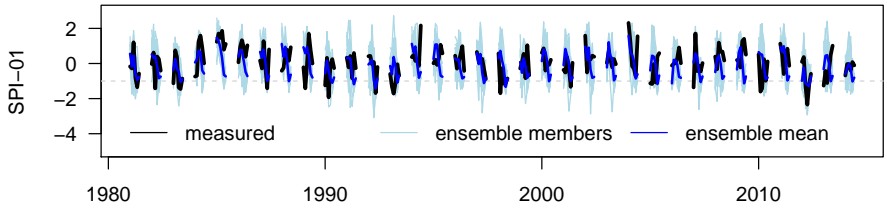

**Figure 6.** Time series of the seasonal forecast of $SPI_1$ in the Castanhão subcatchment given by ECMWF:EQM. Only periods from January to June are shown. The threshold "moderate drought event" is given by the grey dashed line.

observed events of moderate to severe drought (below the dashed grey line). The ensemble mean is able to forecast at least a few of these events.

A measure of the general agreement (for all kinds of conditions, dry, average or wet) between forecasted and observed MDI is given by the RMSE in Fig. 7. The relationship between forecast probability and relative frequency of a drought event (i.e. the

BSS) is provided in Fig. 8, whereas the balance between hit rate and false alarm rate for the same event can be seen in the form of ROCSS in Fig. 9 below.

The RMSE of MDI forecasts is shown in Fig. 7. With the exception of the predictions produced by the WP approach for $SPI_1$, the general ranking of the approaches is quite consistent among the three subbasins. As with precipitation, the RMSE of $SPI_1$ and $SPEI_1$ generally does not differ from that of the climatology and is greatest for ECHAM and EQM.

EQM:ECMWF and XDS:ECMWF show consistently lowest RMSE and XDS:ECMWF performs better than the climatology. Interestingly, ECMWF consistently outperforms ECHAM on all scales.

RMSE reflects the prediction skill for the whole range of the indices, including wet spells and dry spells/droughts. When aiming primarily at forecasting drought events, this verification may be misleading. Nevertheless, this metric shows which models are most appropriate for this domain and confirms the plausibility of the forecasting system also for wet years.

As for the BSS, Fig. 8 shows this indicator of skill for time scales of 1, 3 and 6 months in three regions of the Jaguaribe river. For the one month time scale, it is noteworthy that the first three months of the forecast display the lowest skill. In particular the March forecast shows no skill in most models, March being a key contributing month in the rainy season. The second half of the rainy season, April/May/June, has generally more skill. The same BSS minimum can be observed on the $SPI_3$ and $SPI_6$ panels, but this time with slightly greater value than for $SPI_1$, since these indeces entail some measured data. Another

interesting observation is that, contrary to RMSE, here no product can be considered a clear winner.

For the ECHAM model a possible explanation for lower skill on the onset of the rainy season may lie on its inital conditions. Since the initial conditions for each model run are provided by the output of an AMIP-type run (Gates et al., 1999), they may depart considerably from actual atmospheric conditions. According to this hypothesis, the model would come closer to atmospheric conditions only through the SST forcing, which could explain a certain lag in the forecasting skill. Still, this

explanation can only account for ECHAM and not the ECMWF model, which is fully coupled and whose initial conditions are derived from ERA Interim.

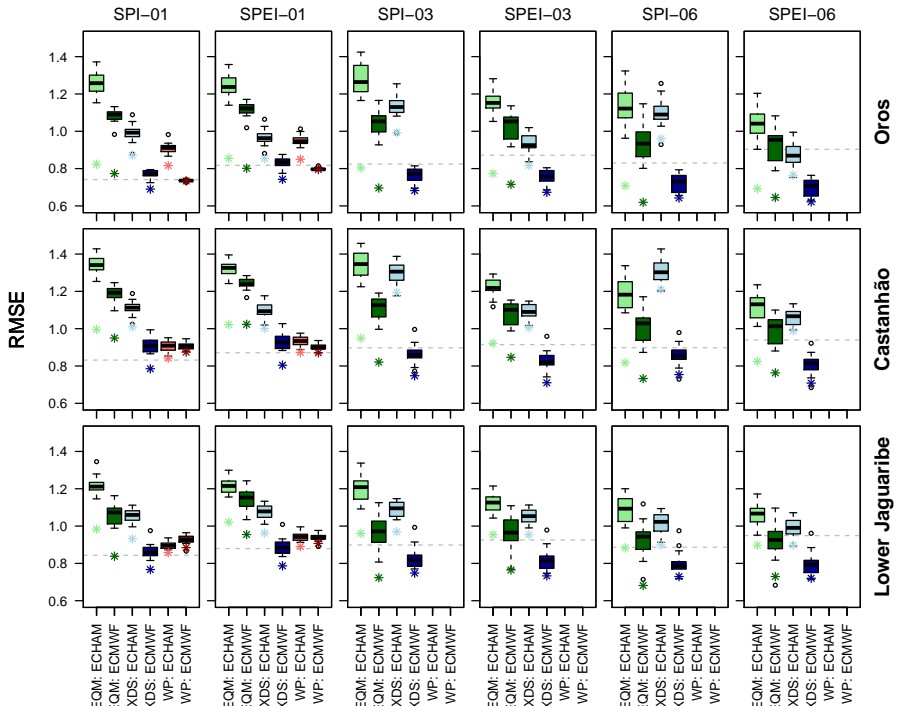

**Figure 7.** box plots of the root mean square error of forecasted meteorological drought index. Asterisk (*) shows the RMSE of the ensemble mean and box plots show the spread of the individual members. Note that in general the ensemble mean ranks better than the best of the ensemble members.

The ROCSS for the different months of the forecasting period shows a slightly different picture than the RMSE and BSS previously presented. Fig. 9 shows ROCSS for time scales of 1, 3 and 6 months in three regions of the Jaguaribe river. There is no clear pattern concerning the relationship between lead time and skill for any of the forecasting models. As in previous plots, the forecasting skill for different MDIs tend to display a minimum in March.

5     Contrary to the results for RMSE, ECHAM shows comparably good ROCSS and BSS in forecasting MDI drought events of all scales in all three regions. Still, the comparably low skill of the March forecast is problematic, March being the month of greatest precipitation in most of the catchment. WP:ECHAM features the best BSSs for $SPI_1/SPEI_1$ in April and May, whereas EQM:ECHAM features generally the highest ROCSS in April for the same scale.

It is worth to look at the BSS of $SPI_6/SPEI_6$, even if they partly encompass measured values. BSS in June in particular

10   is a valuable indicator of the ability of the models in forecasting the whole rainy season. Here, most products display some skill in forecasting a drought event. XDS:ECMWF is the only displaying no skill for all three regions in $SPEI_6$. Generally the skills are higher with $SPEI_6$ than with $SPI_6$. Regarding $SPEI_6$, EQM:ECHAM and EQM:ECMWF display skill in all three regions. In the important region of Castanhão, where the largest reservoir and most infrastructure is located, EQM:ECHAM and XDS:ECHAM perform best in forecasting $SPEI_6$ for June, although with a low value of BSS.

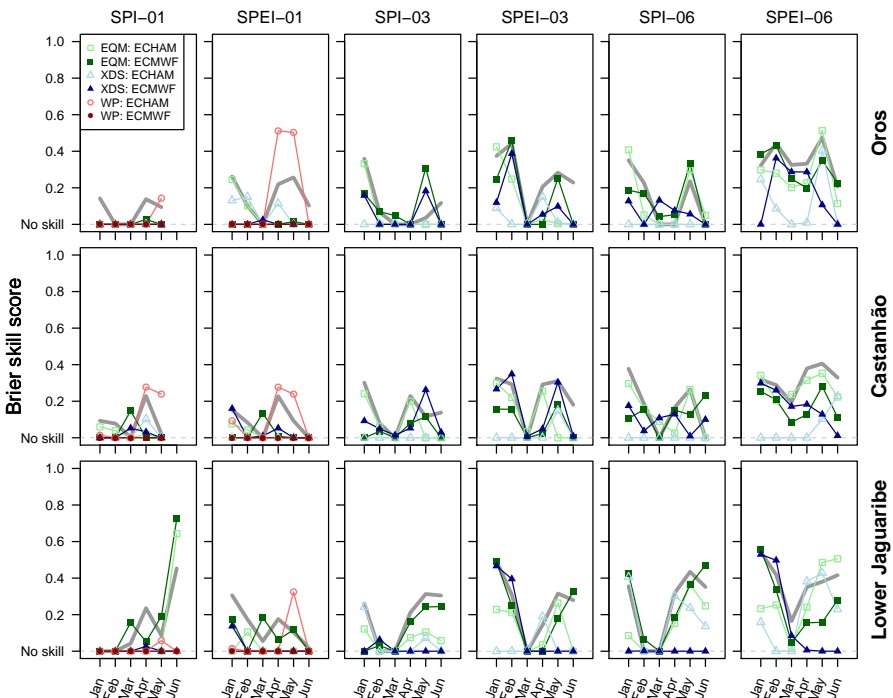

**Figure 8.** BSS of forecasted meteorological drought event based on an event of index lower than -1. The grey line shows the result of the multi-model ensemble mean.

The multi-model ensemble skill, shown by the grey line is generally close to the upper envelope formed by that of the individual models. For $SPEI_1$ in the months January to May (rainy season) the ROCSS of the multi-model ensemble is always positive and oscillates around 0.5. An interesting result is the improvement in skill when $SPI_1$ is replaced by $SPEI_1$. The gray line, which shows the ROCSS/BSS for the multi-model ensemble, has an increase in skill at all scales and regions.

5   A similar forecast assessment has been reported by e.g. Dutra et al. (2013). Events were defined by a $SPI_3$ lower than $-1$, with a lead time of 3 months. ROCSS obtained were in the order of 0.6 for the Blue Nile basin, which is comparable with the results presented in this paper, but much lower for other river basins e.g. Zambezi.

### 3.3   Forecasting hydrological drought indices

The multivariate regression model equations obtained and their respective $R^2$ are shown in the appendix, Table A1. Long
10   scale MDIs (like $SPI_{12}$ or $SPI_{36}$) prevail as predictors of reservoir volume, whereas short scale MDIs like $SPI_1$ are mostly present as predictors of reservoir volume change. This reflects the time scale of reservoir storage variations. At a given moment in time, the reservoir storage results from several months of inflow. Similarly, the effect of a month of high inflow in the reservoir storage level is likely to be only residual.

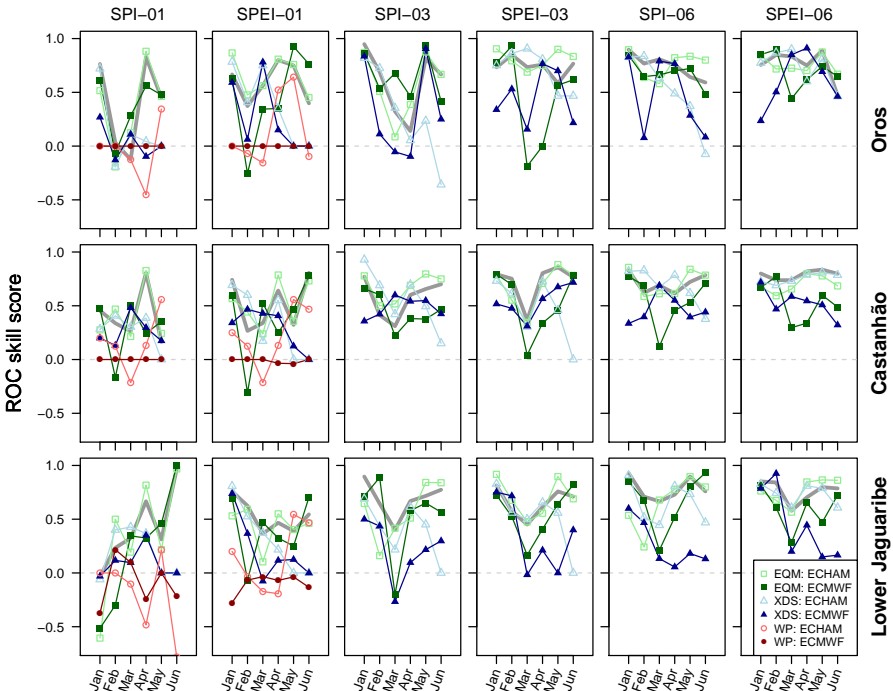

**Figure 9.** ROCSS of forecasted meteorological drought event based on an event of index lower than -1. The grey line shows the result of the multi-model ensemble mean.

The forecast of the three HDIs shows notable differences between downscaling techniques EQM/XDS and the WP classification (Fig. 10). WP classification has lower RMSE than EQM/XDS when predicting SWSI or $V_{M1}$. For $V_{M2}$, the difference between WP and EQM/XDS is much smaller. The ensemble spread of WP classification shrinks considerably from $V_{M1}$ to $V_{M2}$. All methods show a decrease in RMSE from $V_{M1}$ to $V_{M2}$. .

5    Again, WP classification considers by design only a range of discrete MDIs, which can affect RMSE. MDIs were limited to nine values, of which -0.75, 0 and 0.75 are the closest to zero. The continuous values of MDI derived by the other products are problematic, because the multilinear regression also considers division by the meteorological drought index. When the MDI are close to zero, outliers arise and skew the RMSE. These datapoints were therefore removed from the verification metrics.

Regarding the prediction of HDI drought events, Fig. 11 clearly points out that prediction performs best when targeting reservoir volume with model M2 (adding predicted monthly value to the December observed regional reservoir volume). Here, all products show reasonable performance for most regions, but a decreasing skill with increasing lead time. Another important observation is that WPs do not display skill in forecasting HDI events as shown in Fig. 11.

Contrary to the MDIs, the BSS of the HDIs do not feature a minimum in March. A slight tendency of lower skills towards the end of the rainy season is observable in $V_{M1}$ forecasts. $V_{M2}$ shows comparably good results for all GCM/downscaling combinations in predicting HDI events, confirming the results in Fig. 10.

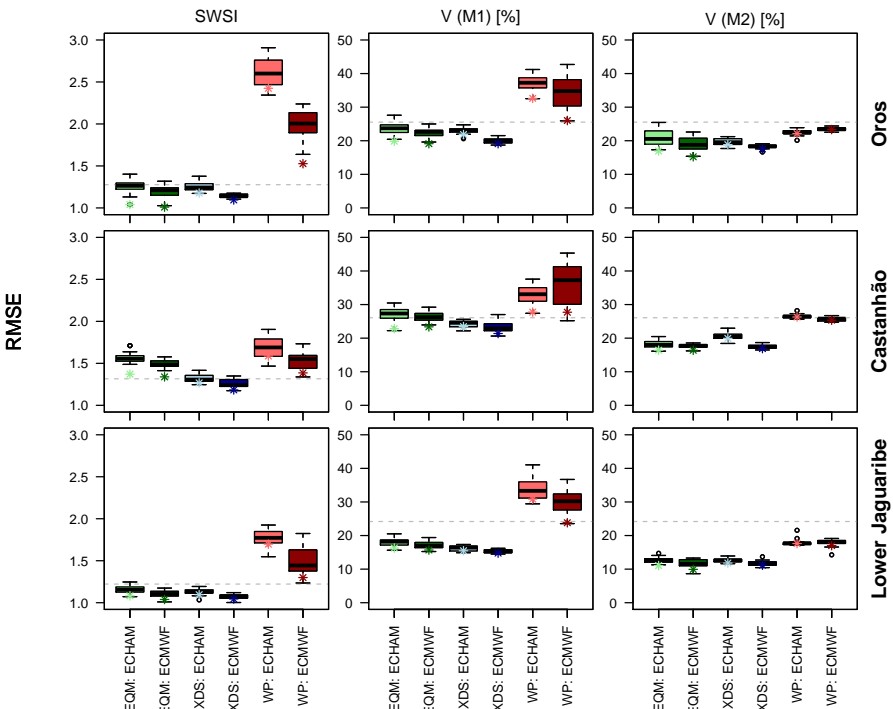

**Figure 10.** Root mean square error of the forecast of SWSI, $V$ predicted with $M1$ and $V$ predicted with $M2$ (based on month-to-month variation). The forecast period is January to June. Three regions are presented: Lower Jaguaribe, Orós and Castanhão. The horizontal grey dashed line shows the RMSE of the climatology.

The ROCSS shows small differences between GCMs or downscaling methods (Fig. 12). $V_{M2}$ features the highest ROCSS of the different indexes used and very little variability among downscaling approaches and GCMs employed. As with BSS, the ROCSS of $V_{M2}$ decreases with increasing lead time. The results of SWSI and $V_{M1}$ are very similar, with SWSI showing higher variability among downscaling approaches and GCMs. $V_{M2}$ could be predicted by WP classification with high ROCSS, whereas $V_{M1}$ and SWSI show no skill.

It was possible to predict any of the indexes with skill in most modelling approaches and catchments. Still, $V_{M2}$ was predicted with the greatest BSS and ROCSS, even though it showed worse $R^2$ when fitting the regression model on which the prediction is based (Table A1). This result hints at better HDI predictability when the predictant is a change in reservoir volume than the reservoir volume itself. One reason for the improved predictability of $V_{M2}$ is surely the importance of persistence in reservoir storage dynamics. By adding the predicted change to the measured reservoir volume we are providing valuable measured information to the forecast model that SWSI and $V_{M1}$ do not have.

We could not find reports on streamflow/reservoir forecasting systems for the region of Ceará stating BSS, ROCSS, RMSE or other verification measure. Still, for other semi-arid regions of the world, similar skill values could be found in the literature. Trambauer et al. (2015) forecasted events of standardized runoff index of 6 months lower than -0.5 with variable lead times.

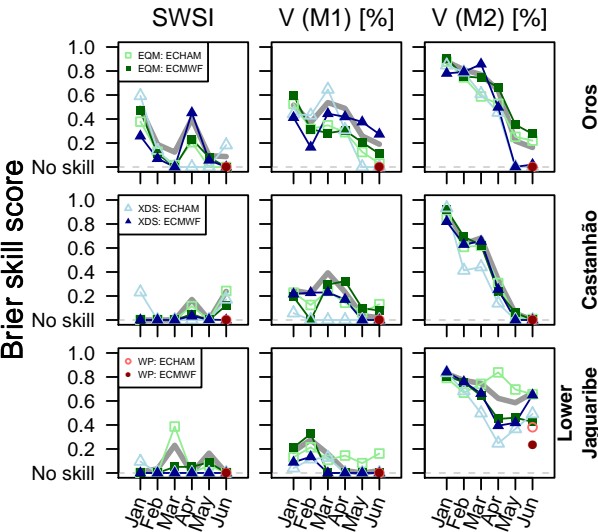

**Figure 11.** BSS of the forecast of drought events as defined by SWSI, $V$ predicted with $M1$ and $V$ predicted with $M2$ (based on month-to-month variation). An event is defined as an index of magnitude lower than the 30th percentile of the observations. The forecast period is January to June. Three regions are presented: Lower Jaguaribe, Orós and Castanhão.

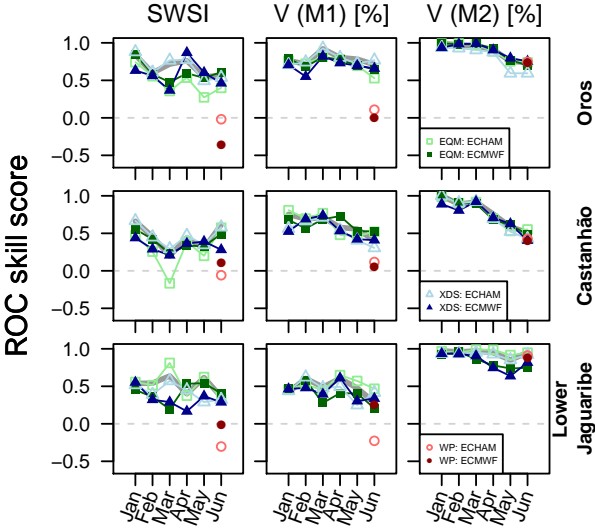

**Figure 12.** ROCSS of the forecast of drought events as defined by SWSI, $V$ predicted with $M1$ and $V$ predicted with $M2$ (based on month-to-month variation). An event is defined as an index of magnitude lower than the 30th percentile of the observations. The forecast period is January to June. Three regions are presented: Lower Jaguaribe, Orós and Castanhão.

**Table 2.** BSS of January-May multi-model ensemble forecast. The ensemble includes ECHAM and the ECMWF seasonal forecast model, as well as the EQM and XDS downscaling techniques. The BSS are averaged over each region. Columns show different indices used for the forecast: P is seasonal precipitation, $SPI_1$ and $SPEI_1$ are standardized precipitation indices with scale 1 month, and *Reservoir Volume* stands for regional reservoir volume in percentage of regional storage capacity. The BSS refers to meteorological and hydrological drought events described in Sect. 2

| region | BSS | | | | | |
|---|---|---|---|---|---|---|
| | P | $SPI_1$ | $SPEI_1$ | $SPEI_6$ | *Reservoir Volume (M1)* | *Reservoir Volume (M2)* |
| Orós | no skill | no skill | 0.11 | 0.38 | 0.43 | 0.66 |
| Castanhão | no skill | 0.08 | 0.11 | 0.32 | 0.23 | 0.52 |
| Lower Jaguaribe | 0.06 | no skill | 0.16 | 0.37 | 0.12 | 0.71 |

Their best catchment point to a ROCSS of 0.7 with a lead time of 5 months. Seibert et al. (2017) forecasted events with a standardized streamflow index (Vicente-Serrano et al., 2012) below -0.5, reporting ROCSS of 0 at the outlet of a large river (the Limpopo in southern Africa) to close to 1 in its headwater catchments.

### 3.4 Multimodel ensemble forecast

Finally, we present the skill score of the multi-model ensemble forecast in Table 2. Each type of index considered (precipitation, meteorological drought index and hydrological drought index) is presented. Results of the WP classification were excluded from the multi-model ensemble, because they did not cover all the indeces addressed in this work.

   The BSS of forecasts of low precipitation events (given in column *P*), as well as that of the forecasts of drought defined by the $SPI_1$ show either very low or no skill. Forecasts of $SPEI$ generally display greater skill than the forecasts of $SPI$. This
points to a possible bias in the forecasting that is compensated by introducing temperature in the equation of SPEI.

   The best skill obtained by the multi-model ensemble was forecasting drought events related to reservoir storage in the Lower Jaguaribe region. The good skill of the reservoir storage forecast is likely related to the long memory of the reservoir system. The forecasted precipitation will affect the reservoir only marginally, since most of its storage is accumulated throughout several years. Most importantly, BSS increases when $V_{M2}$ is used instead of $V_{M1}$, i.e. when reservoir volume is forecasted by
adding forecasted reservoir volume change to measured December reservoir volume.

   Table 2 reveals an interesting pattern in this work: additional information to the forecast model tends to increase forecast skill. $SPEI_1$ is based on temperature and precipitation data and was forecasted with greater skill than $SPI_1$, which is only based on precipitation. Similarly, $SPEI_6$, which combines forecasted and measured precipitation and temperature from months prior to the forecasting period have more skill than $SPEI_1$ forecasts. The greatest BSS is given by $V_{M2}$, a HDI that requires measured
initial reservoir volumes as well as a combination of several MDIs. This last point stresses the importance of assimilating prior hydrological conditions into the forecast products.

## 4  Conclusions

The plausibility and skill of a set of drought forecasting models was presented. Different types of drought events were considered: a rainfall anomaly during the rainy season, standardized precipitation indices below a given threshold and anomalies in regional reservoir storage. The forecast products considered were combinations of two models, ECHAM and the ECMWF seasonal forecast, two downscaling techniques, XDS and EQM, and a weather pattern classification approach.

Each model provided an ensemble of predictions, so deterministic and probabilistic measures of skill could be used. The deterministic measure allowed to see the significant improvement introduced by the ensemble mean: the ensemble mean had in most cases lower root mean square error than the climatology. The RMSE of the ensemble mean however was comparable to the climatology and in some cases greater. Still, no approach had a RMSE that significantly departed from the RMSE of the climatology.

A multi-model ensemble forecast was obtained by binding all members of all models into one product. The skill of this forecast is given in Figs. 4, 8, 11, and Table 2. Multimodel ensembles can be considered to be our best guess of a probabilistic drought forecast, since they are consistently among the best forecast skills provided by the individual models. Individual members surpassed the multi-model ensemble skill only occasionally, for particular combinations of regions, months and indices.

The skill of the hydrological drought forecast, namely the relative reservoir storage $V_{M2}$ was 0.66, 0.52 and 0.71 for the regions of Orós, Castanhão and lower Jaguaribe, respectively. The skill obtained for the hydrological drought forecast is likely inflated by the long memory of the reservoir system and the use of observed reservoir volume to define the conditions prior to each forecast. Still, the $R^2$ of the regression that provides the reservoir variation underlying $V_{M2}$ was lower than that of $V_{M1}$, indicating that a regression might be a poor prediction of reservoir inflow. Improvements are expected by coupling a process-based hydrological model to the seasonal forecasting system.

This work showed that a multi-model ensemble can forecast drought events of time scales relevant to water managers in northeast Brazil. But none or little skill could be found in the forecasts of monthly precipitation or drought indices of smaller temporal scales, like $SPI_1$. Both this work and others here revisited showed that major steps forward are needed in forecasting the rainy season in northeast Brazil.

## Appendix A:  Appendix A

*Author contributions.* Conceived and designed the experiments: TF. Performed the experiments: SV, MR, GB, KV, AM, JD, FVJ, EM. Analyzed the data: JD, SV, TF. Wrote the paper: JD, TF, SV.

*Competing interests.* There are no competing interests present.

**Table A1.** Regression used for predicting regional reservoir volume and regional reservoir volume change using a set of MDIs as predictors. Regional reservoir volume was taken at the end of each month relative to the total regional reservoir storage capacity. Regional reservoir volume change refers to the difference between the given and the previous month.

| Region | Predictant | Formula | $R^2$ |
|---|---|---|---|
| Orós | Reservoir Volume | $59.0 - 22.9SPEI_{36} + 6.67SPI_{12} + 45.4SPI_{36} + 6.00SPI_{12}SPEI_1 - 5.30SPEI_{36}SPI_{36}$ | 0.64 |
| Orós | Reservoir Volume Change | $0.416 + 2.43SPEI_1SPI_1 + 2.23SPI_1SPI_{12} - 0.173SPI_{36}\frac{SPEI_{36}}{SPEI_{12}} + 4.20SPEI_1\frac{SPI_{12}}{SPEI_{12}} - 0.00334\frac{SPEI_{36}}{SPEI_{12}}\frac{SPI_{36}}{SPI_{12}}$ | 0.36 |
| Castanhão | Reservoir Volume | $55.4 + 12.5SPEI_{36} + 12.4\frac{SPI_{36}}{SPEI_{36}} - 3.12SPEI_{12}\frac{SPI_{12}}{SPEI_{36}} - 3.19\frac{SPI_{12}}{SPEI_{36}}\frac{SPI_{12}}{SPEI_{12}} + 10.19\frac{SPI_{36}}{SPEI_{36}}SPEI_{12}$ | 0.41 |
| Castanhão | Reservoir Volume Change | $2.95 + 3.47SPI_1 - 1.15\frac{SPI_1}{SPEI_1} - 1.27\frac{SPI_1}{SPEI_1}SPEI_{36} - 0.791SPI_1\frac{SPI_{36}}{SPEI_{12}} + 1.41SPEI_{12}\frac{SPI_{36}}{SPEI_{36}}$ | 0.21 |
| Lower Jaguaribe | Reservoir Volume | $33.4 + 16.3SPEI_{36} + 16.6SPI_{12} + 5.65SPEI_{36}SPI_{12} - 13.5SPI_{12}\frac{SPI_{12}}{SPEI_{12}} + 0.877SPI_{12}\frac{SPI_{12}}{SPEI_{36}}$ | 0.60 |
| Lower Jaguaribe | Reservoir Volume Change | $0.689 + 2.22SPI_1 + 0.0353\frac{SPI_{36}}{SPI_{12}} + 2.12SPI_1SPI_{12} + 1.08SPI_{12}\frac{SPI_{12}}{SPEI_{12}} + 0.286SPI_1\frac{SPI_{12}}{SPEI_{36}}$ | 0.38 |

*Acknowledgements.* This work was funded by the Federal Ministry of Education and Research of Germany under grant number *01DN14013*. The first author was also supported by the German Research Foundation under grant number *BR1731/18-1*. One of the hindcast datasets was kindly provided by the European Centre for Medium-Range Weather Forecasts. The Open Access Publication Fund of the University of Potsdam supported the publication of this research paper.

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
