# Peer review of "Seasonal Drought Prediction for Semiarid Northeast Brazil: Verification of Six Hydro-Meteorological Forecast Products"

_Hydrology and Earth System Sciences, 2017_

## Referee Comment (RC1) · Anonymous Referee #1 · 18 Nov 2017

The manuscript describes verification of seasonal drought prediction frameworks using six hydro-meteorological data sets as input. Overall the manuscript is well-written and potentially interesting. However, the methodology section is not well explained. Moreover, the verification scores used in the manuscript couldn't evaluate the performance of the frameworks, comprehensively. It suffers from some drawbacks listed in "specific comments" and "technical corrections".

specific comments

1- Reasons for choosing the two specific GCM models, ECMWF and ECHAM, and the manner in which output of the models were combined needs further description to

enhance readability of the manuscript.

2- Further description on calculation of SWSI should be added to the manuscript.

3-It is not fully cleared that why only time scales of 1, 12 and 36 months are selected for drought indices. It is clear that long time scales would be more meaningful in deriving reservoir storage by a regression equation. However, using time scales longer than 6 months distorted evaluation of the forecasting framework.

4-RMSE, used in the manuscript, measures "average" error, weighted according to the square of the error and does not indicate the direction of the deviations. Moreover, ROC skill score used for evaluation of the proposed prediction framework. One of key features of any prediction framework is to evaluate reliability of the predictions. Reliability is the average agreement between the forecast values and the observed values. If all forecasts are considered together (not categorized into bins), then the overall reliability term is used which is the same as the bias.

Generally, ROC skill score is insensitive to forecast biases. Thus, care is needed when interpreting and comparing the performance of model forecasts based on the ROC score. It is recommended to use other skill scores reflecting reliability (bias) in addition to ROC skill score. Brier Score may be a choice.

5- On page 15, line 27: "Results showed that models with little to no skill by a deterministic measure (RMSE) showed skill under a probabilistic skill score. This underlines the importance of having at least one deterministic and one probabilistic measure of skill."

RMSE and ROCSS measures different aspects in evaluation of a prediction framework. In other words, different values for those scores are because they measure different attributes. So, such a conclusion is weak.

6- Justification needed on choosing -1 as threshold of the drought indices for drought events.

Interactive
comment

7- In regression, do you have any suggestion to address the problem of division by indices when the indices are close to zero?

technical corrections

1- On page 5, line 18, citation of "Philipp et al. (2016)" should be modified to the correct form: "Philipp et al. (2014) ".

2- On page 5, line 24, change "The numbers in the index name refer" to "subscripted number refers"

3- The table on page 14 doesn't have any caption.

4- In whole text, "Sect." should be written in its full form: "section"

5- Gray solid line indicating values of multi-model ensemble should be mentioned in legends of Figs. 4, 7 and 9.

6- Fig. 7 is not showing boxplots. Change the caption.

---

## Referee Comment (RC2) · R. Mirabbasi (Referee) · 10 Mar 2018

Dear Prof. Carlo De Michele, As desired, my comments on the manuscript entitled " Seasonal Drought Prediction for Semiarid Northeast Brazil: Verififlfication of Six Hydro-Meteorological Forecast Products" (Ms. No. HESS-2017-572) are listed below. In my point of view, this paper can be accepted after revision.

With kind regards, Rasoul Mirabbasi

General comment The manuscript is concerned with assessing the plausibility and skill of a set of drought forecasting models. The considered forecast products were combinations of two models, ECHAM and the ECMWF seasonal forecast, two downscaling techniques, XDS and EQM, and a weather pattern classiï̈ňȦcation approach. In my personal opinion, the present paper fits in the broad scope of the journal. However, it is necessary to modify the present paper before publication. Questions, suggestions and comments regarding the manuscript are listed below:

Major Comments - The methodology is presented very brief and in some cases vague (e.g. Multimodel ensemble forecast). It suggested to explain more about the applied methods. - In Table 2, the authors presented some regression equations for predicting HDI from MDIs. When they stated that forecasting of MDIs haven't good accuracy (specially when increase the lead time), how they expected to achieve acceptable results in calculating HDI from MDIs? Did the author consider the uncertainties in predicting the HDI? - In Figure 5, the authors considered -1 as the threshold for capturing dry spell. As they stated in the manuscript (Page 8, line 9) that in the study area the January, May and June are months with low or no precipitation and February and March are months with high precipitation, therefore the SPI and SPEI cannot account for seasonal variability, i.e. a given amount of precipitation has different effects on moisture status depending on when it occurs. So it suggests to consider the monthly mean as threshold instead of overall mean and calculate the modified SPI and SPEI (See Kao and Govindaraju, 2010; Mirabbasi et al., 2013 for more information). - Kao, S.C., Govindaraju, R.S., 2010. A copula-based joint deficit index for droughts. J. Hydrol. 380, 121-134. doi:10.1016/j.jhydrol.2009.10.029

- Mirabbasi, R., Anagnostou, E.N., Fakheri-Fard, A., Dinpashoh, Y., & Eslamian, S. (2013). Analysis of Meteorological Drought in Northwest Iran using the Joint Deficit Index. Journal of Hydrology, 492: 35-48 DOI: 10.1016/j.jhydrol.2013.04.019.

- The authors should explain why they choose the 1-, 12- and 36- months time scales for computing the drought indices (MDIs)? As they stated in the manuscript, by increasing the lead time, the accuracy of forecasting drought decreased. What are the advantages of forecasting drought by 36 months time scale? Also the authors said that

for computing the SPI36/SPEI36 forecasts most of the contributing months are coming from past observations and not within the forecast period. So they cannot make correct judgment about the skill of the models in forecasting drought indices with long-term time scales. Therefore, I think calculating the drought indices with smaller time scales (such as 3, 6, 9 months) and higher accuracy will be more acceptable and useful for farmers or water resources managers. Minor Comments - The authors should explain in the manuscript why the seasonal forecasting model only runs for 8 months not for all months of the year? When you consider only 8 months, How you calculate the drought indices with 12 or 36 months time scale? - On page 6, Line 9, (and Page 7, line 18) the authors choose the 30th percentile of each index as the threshold for capturing dry spells. Why you choose this threshold? - On page 7, line 13, the phrase "precipitation in he Jaguaribe basin" should be corrected as "precipitation in the Jaguaribe basin"

———————————————————

---

## Author Comment (AC1) · 6 Apr 2018

We would like to thank the detailed and constructive comments of referee #1. Generally we agree with the technical corrections and specific comments. In particular the methods section, which was also mentioned as too short by the second referee, will be extended to accommodate all the additional information mentioned in the review process.

1 - Reasons for choosing ECMWF and ECHAM models

The main reasons were the availability of the models both for the hindcast period and

in operation. We shall further explain the reasons for the choice and processing of their output in the revised manuscript.

2 - We will explicitly show the equation describing SWSI in the revised manuscript.

3 - The reason for using longer time scales is the nature of droughts in northeast Brazil: their scale is often interannual, if not decadal. By using longer time-scales the forecast of the current rainy season is put into the context of an interannual drought. As you correctly state, although useful for forecasting reservoir levels, SPI36 is not very useful for verification. We will therefore focus in the verification of shorter SPIs in the revised manuscript.

4 - We agree that a measure of reliability should be shown. We will try to arrange the manuscript in order to fit reliability plots and Brier score in it without extending it too much,

5 - You are right, RMSE and ROCSS measure different aspects, which happen to be related to deterministic and probabilistic forecasts. Our conclusion is misleading and we will reformulate it.

6 - We will add an explanation for choosing -1 as a threshold for drought.

7 - Although the solution is not very elegant, we had to truncate the indeces within ]-0.1, 0] and [0, 0.1[ to -0.1 and 0.1 respectively. We will describe the method in the revised manuscript.

---

## Author Comment (AC2) · 6 Apr 2018

We would like to thank Prof. Rasoul Mirabbasi for the constructive comments to our manuscript. We identified 4 major comments that we will address in this reply.

1 - Extent of the methodology section

We agree that the methodology section is too short and should be extended. In the revised manuscript we will extend the description of the model ensemble, as well as the formulation of the different indexes employed in the paper.

2 - The uncertainty of the regression of HDI based on MDI

[Figure]

This remark is in part related to the previous one. The presentation of the MDI/HDI regression is extremely short in the submitted manuscript. The same comment applies to the discussion of the results of the regression. We shall extend this subsection, in particular discussing the explained variance in the regression of each index and region.

3 - Influence of strong seasonality in the MDI and using the joint deficit index

Although in principle we agree that drought is a multiscale phenomenon – and that Kao and Govindaraju (2010) and Mirabbasi et al. (2013) provide an interesting solution for quantifying multiscale drought severity – we don't see their method appropriate in the context of this paper. The aim of this manuscript was to provide a clear and standard verification of the forecast models that can be recognized as such by the community. By introducing relatively new and specific concepts in the manuscript we would be re-centering the whole scope of our work. As mentioned in the reply to the anonymous referee #1, we will use shorter time scales for the verification in the revised manuscript.

4 - Usefulness of long time scales of MDI

We agree with your comment: concerning the verification, it is more useful to consider shorter time scales of the MDIs, eg for small hold farmers. However, the scale of droughts in northeast Brazil is often interannual, if not decadal. By using longer time scales the forecast of the current rainy season is put into the context of an interannual drought. Furthermore, longer time scales proved to be better predictors of reservoir volume.

In the revised manuscript, we will use shorter time scale of the MDIs in the verification of the forecast models, but keep longer time scales in our manuscript, since they provide better predictors for reservoir volume and useful information for managers and decision makers.

---

## Author Response (AR1)

Dear Prof. De Michele,

I hereby resubmit the manuscript hess-2017-572, with first submission on 21 Sep 2017:

> Seasonal Drought Prediction for Semiarid Northeast Brazil: Verification of Six Hydro-Meteorological Forecast Products

We introduced some major revisions addressing the requests of the referees:

**1 The methods section was expanded**
**2 Brier skill score was added to the verification (Figs. 4, 8 and 11 in the revised manuscript)**
**3 We further explained the reasons for using the ECHAM and ECMWF models**
**4 Focus of the manuscript moved from longer to shorter time scales of the meteorological drought indeces (1, 3 and 6 months)**
**5 A new paragraph explains the choice of drought thresholds**
**6 Methods section was expanded regarding SWSI including its formal expression**
**7 Methods section was expanded to explain the truncation of index values close to zero when deriving HDIs**
**8 Conclusions were reformulated and no longer compare ROCSS and RMSE**

Another major revision, not requested by the referees, was removing the verification of the forecasted reservoir volume variation. Instead, we added that same forecast (months January to June) to the measured December reservoir volumes. This way we obtained a forecasted reservoir volume, comparable to the direct regression of the reservoir volume of MDIs. The methodology is explained in the reviewed manuscript.

Other revisions:

**improved color legend of figures 3 and 5 (revised manuscript)**
**moved Table 2 (first version of the manuscript) to the appendix**

Minor revisions to the text are marked in the attached document and listed in pages 27 to 31.

Best regards,
Jose Miguel Delgado

**Seasonal Drought Prediction for Semiarid Northeast Brazil: Verification of Six Hydro-Meteorological Forecast Products**

José Miguel Delgado[1], Sebastian Voss[1], Gerd Bürger[1], Klaus Vormoor[1], Aline Murawski[2], José Marcelo Rodrigues Pereira[3], Eduardo Martins[3], Francisco Vasconcelos Júnior[3], and Till Francke[1]

[1]Institute of Earth and Environmental Sciences, University of Potsdam, Germany
[2]German Research Centre of Geosciences GFZ Potsdam, Germany
[3]Research Institute for Meteorology and Water Resources - FUNCEME, Fortaleza, Brazil

*Correspondence to:* José Miguel Delgado (martinsd@uni-potsdam.de)

**Abstract.** A set of seasonal drought forecast models was assessed and verified for the Jaguaribe River in semiarid northeast Brazil. Meteorological seasonal forecasts were provided by the operational forecasting system used at FUNCEME (Ceará's research foundation for meteorology) and by the European Centre for Medium-Range Weather Forecasts (ECMWF). Three downscaling approaches (empirical quantile mapping, extended downscaling and weather pattern classification)[JMD] were tested and combined with the models in hindcast mode for the period 1981 to 2014. The forecast issue time was January and the forecast period was January to June. Hydrological drought indices were obtained by fitting a multivariate linear regression[JMD1] to observations. In short, it was possible to obtain forecasts for *a)* monthly precipitation, *b)* meteorological drought indices, and *c)* hydrological drought indices.

The skill of the forecasting systems was evaluated with regard to root mean square error (RMSE), the Brier skill score (BSS)[JMD] and the relative operating characteristic skill score (ROCSS)[JMD]. The tested forecasting products showed similar performance in the analyzed metrics.[JMD] Forecasts of monthly precipitation had little or no skill considering RMSE and mostly no skill with BSS[JMD] [JMD]. A similar picture was seen when forecasting meteorological drought indices: low skill regarding RMSE and BSS[JMD] and significant skill when discriminating hit rate and false alarm rate given by the ROCSS (forecasting drought events of e.g. $SPEI_{01}$ showed a ROCSS of around 0.5)[JMD]. [JMD] Regarding the temporal variation of the forecast skill of the meteorological indeces[JMD], it was greatesr[JMD] for April, when compared to [JMD]the remaining months of the rainy season)[JMD],
[revised manuscript text omitted]

* * *
[9]JMD: replacing use of SPI12

[10]JMD: clarifying use of long scales in HDI

[12]JMD: introducing swsi

[13]JMD: introducing V

In terms of event prediction, the event considered for the meteorological drought indeces in use in this work is "dry spells of moderate to extreme magnitude", translated by values lower than or equal to $-1$ in the SPI/SPEI scale. For precipitation a threshold based on the 30th percentile of the series of observed monthly precipitation was used. The threshold for defining HDI drought events was based on the 30th percentile of the series of observed monthly HDI. The reason for using the 30th percentile was the classification used by the regional agencies to separate between a "dry", a "wet" (above the 70th percentile) and a "normal" year. [JMD14]

**2.6  Regression of Hydrological Drought indices**

Forecasts of hydrological drought indices were obtained by searching and fitting a multivariate regression model[JMD] to observations of hydrological drought indices and reservoir volume changes. [JMD]As candidate predictors, meteorological drought indeces of all temporal scales were used.[JMD15].

[revised manuscript text omitted]

This work showed that a multi-model ensemble can forecast drought events of time scales relevant to water managers in northeast Brazil with skill. But none or little skill could be found in the forecasts [JMD] of monthly precipitation
* * *
[43] JMD: this is wrong, as the reviewer correctly said

[44] JMD: in fact the opposite is true, I suppose this was a lapse

[45] JMD: same as previous remark, probably a lapse

[46] JMD: improves clarity

[47] JMD: not true with M2

[48] JMD: refer to regression

[revised manuscript text omitted]

**Changes**

**Author: anonymous**

Added . . . . . . . . . . . . . . 0

Deleted . . . . . . . . . . . . . 0

Replaced . . . . . . . . . . . . 0

**Author: JMD (JM Delgado)**

Added . . . . . . . . . . . . . 53

Deleted . . . . . . . . . . . 33

Replaced . . . . . . . . . . . 56

**List of changes**

---

## Referee Report (RR1)

The manuscript describes verification of seasonal drought prediction frameworks using six hydro-meteorological data sets as input. Overall the manuscript is well-written and potentially interesting. However, it still suffers from some drawbacks listed in "specific comments" and "technical corrections".

Specific comments:

1- According to my previous recommendations, Brier Skill score used as a metric to evaluate the performance of forecasts of the model. Brier score is introduced by G. W. Brier in 1950 for probabilistic forecasts. Unfortunately, the variables in equation 2 ($y_k$ and $o_k$) are not fully explained in the manuscript. Theoretically, the forecast probability can adopt any value in [0,1]. However, it seems that they take only values 0 or 1 in your case (dichotomous forecasts). Further explanation is needed. Original (not simplified) formulation of Brier score could be found in:

Bateni, M. M., Behmanesh, J., Bazrafshan, J., Rezaie, H., & De Michele, C. (2018). Simple Short-Term Probabilistic Drought Prediction Using Mediterranean Teleconnection Information. *Water Resources Management*, 1-14.

2- Further explanation is needed in the manuscript for calculation of Brier score of the reference forecast for the desired time period.

Technical corrections:

1- Figs. 11 and 12 are not about RMSE. Please correct the legends of those figures.
2- On page 5, line 26: "The numbers in the index name refer" should be changed to "The subscripted numbers"
3- Abbreviations like BSS or ROCSS should be defined at first mention and used consistently thereafter. In whole text, "Sect." should be written in its full form.
4- Subscripts should be used consistently (not somewhere $SPI_1$ and elsewhere $SPI_{01}$). Do use italics based on guidelines for the journal to avoid ambiguity.

---

## Author Response (AR2)

Dear Prof. De Michele,

I hereby resubmit the manuscript hess-2017-572, with first submission on 21 Sep 2017:

Seasonal Drought Prediction for Semiarid Northeast Brazil: Verification of Six Hydro-Meteorological Forecast Products

The manuscript was subject only to minor revisions, following the indications by the reviewers. We expanded the explanation of the Brier score as well as the definition of the reference forecast with respect to the Brier score in page 8, lines 20 to 29. Furthermore, we corrected:

- the captions of Figs. 11 and 12
- page 5, line 26 (of the previous version)
- inconsistent abbreviations
- inconsistent subscripts

We did not replace "Sect." by "Section" in the text since HESS's manuscript preparation guidelines advise against this.

We would like to thank you for your support during the whole process.

Best regards,
Jose Miguel Delgado